# Simplified Regional Prediction Model of Long-Term Trend for Critical Frequency of Ionospheric F2 Region over East Asia

**Jian Wang** [1,2,*] , **Hongmei Bai** [1,3], **Xiangdong Huang** [2,4,*] , **Yuebin Cao** [2,4], **Qiang Chen** [2,4] **and Jianguo Ma** [2,5]

1   School of Microelectronics, Tianjin University, Tianjin 300072, China
2   Qingdao Institute for Ocean Technology, Tianjin University, Qingdao 266100, China
3   School of Mathematics and Statistics, Hulunbuir College, Hulunbuir 021000, China
4   School of Electrical and Information Engineering, Tianjin University, Tianjin 300072, China
5   School of Computer, Guangdong University of Technology, Guangzhou 510006, China
*   Correspondence: 1016204014@tju.edu.cn (J.W.); xdhuang@tju.edu.cn (X.H.)

**Abstract:** To improve the accuracy of predictions and simplify the difficulty with the algorithm, a simplified empirical model is proposed in developing a long-term predictive approach in determining the ionosphere's F2-layer critical frequency ($f_oF_2$). The main distinctive features introduced in this model are: (1) Its vertical incidence sounding data, which were obtained from 18 ionosonde stations in east Asia between 1949 and 2017, used in reconstructing the model and verification; (2) the use of second-order polynomial and triangle harmonic functions, instead of linear ones, to obtain the relationship between the seasonal vs. solar-cycle variations of $f_oF_2$ and solar activity parameters; (3) the flux of solar radio waves at 10.7 cm and sunspot number are together introduced in reconstructing the temporal characteristics of $f_oF_2$; and (4) the use of the geomagnetic dip coordinates rather than geographic coordinates in reconstructing the spatial characteristics of $f_oF_2$. The statistical results reveal that $f_oF_2$ values calculated from the proposed model agree well with the trend in the monthly median statistical characteristics obtained from measurements. The results are better than those obtained from the International Reference Ionosphere model using both the CCIR and URSI coefficients. Furthermore, the proposed model has enabled some useful guidelines to be established for a more complete and accurate Asia regional or global model in the future.

**Keywords:** ionosphere; F2-layer critical frequency; simplified regional model; long-term prediction; east Asia

## 1. Introduction

The critical frequency of the F2 layer of the ionosphere (or known as plasma frequency of the F2 layer, identified as $f_oF_2$) is a very important characteristic parameter in various civil and military applications such as high-frequency communications [1], satellite communications [2], navigation [3], timing, radar tracking, detections, locations, and spectrum management [4]. The accuracy in predicting $f_oF_2$ is an important way to improve the efficiency for the above- mentioned electronics-based information systems. For example, it can be used to improve the predictions of the receiving area, usable frequency and the group-delay, distribution and therefore help to determine the suitability of the communication [1]. The critical frequency can be measured using vertical [5,6] and oblique incidence sounding ionosonde [7], global navigation satellite systems.

In the absence of real-time measurements, the ionospheric models play an important role in all parts of related research fields. For this reason, many empirical models were developed in the past

to predict $f_oF_2$ [8]. Among these models, some provide global coverage whereas some are regional. The models describing the temporal and spatial characteristics of the ionosphere are divided into two main categories: (1) Theoretical or physical, and (2) mathematical or empirical and can be derived from ionosonde data. There are various empirical ionosphere models to obtain $f_oF_2$ [4,9,10]. Moreover, these models provide long-term predictions of the monthly median values of the $f_oF_2$ and have been used to a certain degree as reference quiet time values [8,11], for both global and regional models.

As is well-known, the International Reference Ionosphere (IRI) model is the most representative global model. Initially, Jones and Gallet [12,13] proposed a global model that expressed the measurements for each station as a Fourier series with special Legendre functions dependent on latitude and longitude for each Fourier coefficient. This model was later adopted for international use by the International Radio Consultative Committee (CCIR) and recommended as International Telecommunications Union (ITU) standard [14]. On the basis of this global model, the IRI was proposed by a joint working group of the CCIR and the International Union of Radio Science (URSI) [15]. The IRI model is currently the most popular model and has been steadily developed and improved with newer data and more advanced techniques by the joint working group of the Committee on Space Research (COSPAR) and URSI. The IRI model has evolved to a number of important editions including IRI-78 [16], IRI-1990 [15], IRI-2000 [17], IRI-2007 [18], IRI-2012 [19], and IRI-2016 [20]. As the latest version, IRI-2016 includes significant improvements not only to the representation of the electron density, but also for the description of the electron temperature and ion composition [21]. In recent years, neural-network (NN) based global models for $f_oF_2$ are being developed and evaluated as a possible replacement for the IRI models presently in modeling [17,22]. Ercha [23] proposed a model for ionospheric $f_oF_2$ that uses an empirical orthogonal function (EOF) analysis. At the same time, each developed model is compared with the IRI model to ensure that the replacement solution has greater accuracy than the IRI solution [17]. The purpose in all the above research is to achieve more accurate prediction of the parameters by developing new models and optimizing the existing models.

Compared with global models, regional models give in general better results with better agreement with observations [8,24,25]. The development of these regional models arose: (1) With the demand for improved performances for specific areas, (2) in response to the availability of denser network of stations, and (3) to simplify the complex ionospheric morphology over a restricted area [25]. Thus, related work has also been continuing in more regional refined studies, such as the simplified ionospheric regional model (SIRM) [26] and its improved version [25] over Europe, the regional ionospheric model over Antarctica [27], the regional ionospheric model over India [28], the maps over northern China [24], as well as others [29]. The accuracy of the derived parameters in above-mentioned applications directly depends on the accuracy of $f_oF_2$ [4]. Therefore, the empirical ionospheric model has been continually improved and further developed by different groups in the public domain making use of the latest measured data [8,19] or introducing advanced techniques, for instance, artificial neural networks [22] and EOF analyses [23,30,31].

A clear trend is evident in focusing on regional models, rather than global models during the past decades, because of their capability to produce a more accurate ionospheric representation over particular areas giving better results for both telecommunications and geophysical modeling [25,32,33]. The ionosphere over East Asia is an important region where the atmospheric dynamics within the high, middle, and low-latitude ionosphere, can be very complicated. Several studies have shown that there are relatively large discrepancies in the values of the ionospheric parameters predicted by the IRI model and the observational data in equatorial and low-latitude regions, especially in East Asia and areas of southern China [23,34]. Therefore, this paper focuses on reconstructing a simplified regional long-term prediction model for $f_oF_2$ over East Asia. In addition, an important objective in this modeling is to simplify the algorithmic complexity and to improve model efficiency based on the basis of guaranteeing accuracy. In this paper, the proposed model is represented as a numerical mapping using the orthogonal temporal and spatial functions. Most importantly, the temporal numerical map functions are reconstructed through two solar activity parameters (F10.7 and sunspot number), and the

spatial numerical map functions are reconstructed by the modified surface interpolation method based on the geomagnetic function instead of the simple geographic coordinates.

In the following sections, we will describe the modeling approach in Section 2, and then the model reconstructed process in Section 3. In the end, the predicted values of the new model will be compared with the measured values and IRI model to validate its effectiveness.

## 2. Modeling Method

### 2.1. Base Algorithm

Our modeling of $f_oF_2$ is based on the EOF and combines harmonic function theory and two-fold regression analysis. This EOF analysis method was proposed by Pearson [35] and first introduced into empirical modeling of the ionospheric parameters by Dvinskikh [36]. It involves a mathematical procedure that transforms a dataset into a number of uncorrelated components called principal components, any two components of which are orthogonal to each other [23]. The EOF analysis method not only considerably reduces the number of modeling parameters but also provides significant rapid convergence with high calculation accuracy. Compared with other methods of expansion, the EOF model provides optimum convergence in the representation of ionospheric characteristics [36,37]. The underlying physical meaning is that the variation of the physical field variable is mainly controlled by some independent processes that can be separated [34]. Therefore, we separated $f_oF_2$ into temporal and spatial variables. According to the above-mentioned theory and analysis, the $f_oF_2$ characteristics can be retained from a numerical mapping technique based on the orthogonal temporal and spatial functions. Specifically, the $f_oF_2$ numerical map function is simply modeled as an orthogonal harmonic function series:

$$f_oF_2(\lambda, \varphi, F_{12}, R_{12}, m, UT) = \sum_{n=1}^{N} w_n(\lambda, \varphi) \cdot \hat{f}_oF_2(F_{12}, R_{12}, m, UT), \tag{1}$$

where $\lambda$ is the geographic latitude, $\varphi$ is the geographic longitude, $F_{12}$ is the twelve-month running mean value of the monthly flux of solar radio waves at 10.7 cm, $R_{12}$ is the twelve-month running mean value of the monthly sunspot numbers, $m$ is the month, $UT$ is the given universal time, $N$ is the maximum number of measured stations used for interpolation, $\hat{f}_oF_2(F_{12}, R_{12}, m, UT)$ is the temporal reconstruction function and is determined in Section 3.1, $W = [w_1, w_2, \ldots, w_N]$ are the weights calculated by the spatial reconstruction equations in Section 3.2.

The flow chart of the model reconstruction and verification is shown in Figure 1. In the following sections, the new modeling approach and the comparison between measured value and different models will be described and analyzed.

### 2.2. Model Verification

In order to evaluate the $f_oF_2$ accuracy of the proposed model, the predicted value and the measured value will be compared and analyzed. The following parameters were calculated:

(**a**) Root-mean-square error (RMSE)

$$\sigma = \sqrt{\frac{1}{H}\sum_{h=1}^{H} \left(f_p{}^h - f_m{}^h\right)^2}, \tag{2}$$

(**b**) Relative root-mean-square error (RRMSE)

$$\delta = \sqrt{\frac{1}{H}\sum_{h=1}^{H} \left(\frac{f_p{}^h - f_m{}^h}{f_m{}^h}\right)^2}, \tag{3}$$

where $f_p$ is predicted value of $f_oF_2$, $f_m$ is measured value of $f_oF_2$, $N$ is the statistics number of data points.

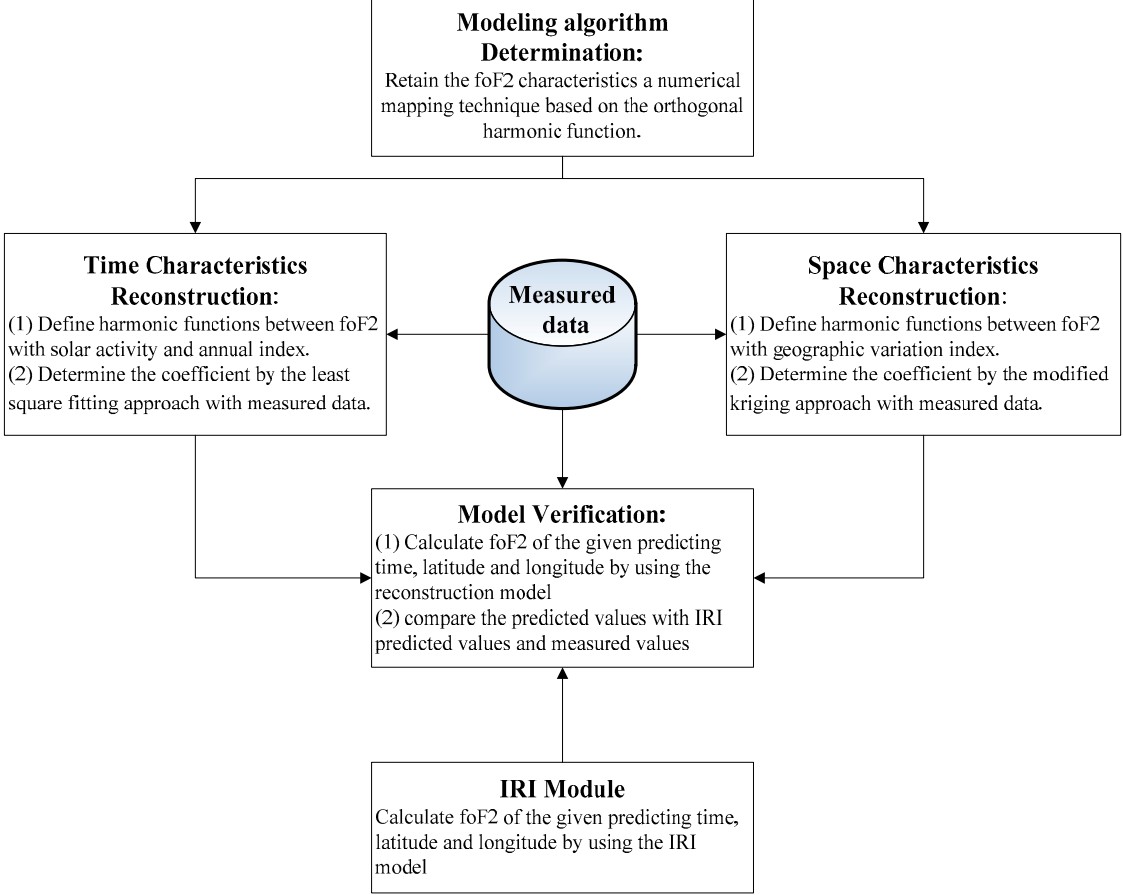

**Figure 1.** The flow chart of the model reconstruction and verification.

*2.3. Dataset*

In this study, daily hourly measured values of $f_oF_2$ and its monthly median values have been obtained from eighteen ionosonde stations in east Asia as shown in Figure 2 and Table 1. These stations include Akita, Beijing, Chongqing, Guangzhou, Haikou, Irkutsk, Jeju, Khabarovsk, Magadan, Manzhouli, Okinawa, Seoul, Taipei, Tokyo, Tunguska, Wakkanai, Yakutsk, and Yamagawa, which used for model reconstruction and verification. The geographical coordinates and geomagnetic latitudes of ionosonde stations are listed in Table 1 and are shown in Figure 2.

Measured data comes from various resources of National Oceanic and Atmospheric Administration (NOAA) and World Data Centre (WDC). The data source includes four parts: (1) The ionosonde data of Akita, Okinawa, Kokubunji, Wakkanai, and Yamagawa station was downloaded from the World Data Centre website (http://wdc.nict.go.jp/IONO/HP2009/ISDJ/manual_txt-E.html); (2) the ionosonde data covering the years of 1957- 2012 of Khabarovsk station was downloaded from the World Data Centers in Russia and Ukraine (http://www.wdcb.ru/stp/data/ionosphere, ionosphere_1, ionosphere_2, ionosphere_4); (3) the ionosonde data covering the years of 1950–2008 of Irkutsk, Seoul, Taipei, Manila, Beijing, Chongqing, Guangzhou, Haikou, and Manzhouli station was downloaded from Australian government-Bureau of Meteorology Space Weather service (ftp://ftp-out.sws.bom.gov.au/wdc/iondata/medians); (4) Other ionosonde data was downloaded from National Oceanic and Atmospheric Administration website (ftp://ftp.swpc.noaa.gov/pub/lists/iono_month). The total data volume of $f_oF_2$ monthly median values is 168674 entries, and that of each station is listed in Table 1.

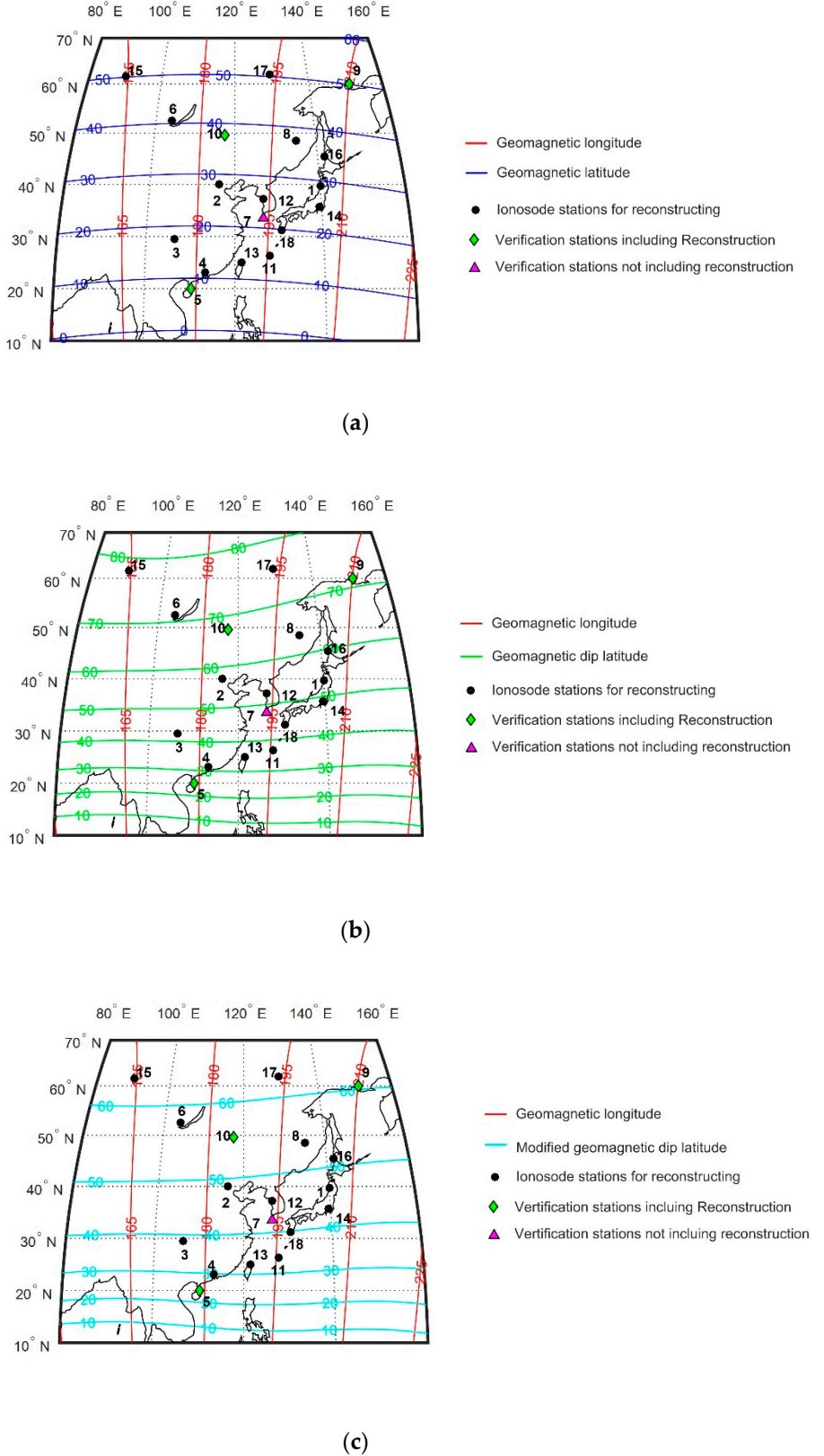

**Figure 2.** Comparison of geomagnetic and geographic coordinates in the analysis region. (**a**) Geographic coordinates and geomagnetic coordinates, (**b**) geographic coordinates and geomagnetic dip latitudes, and (**c**) geographic coordinates and modified geomagnetic dip latitudes.



**Table 1.** Geographic and geomagnetic positions of the ionosonde stations used for model reconstruction and verification.

| Station Label | Station Name | Geog. Lon. (ºE) | Geog. Lat. (ºN) | Geomag. Lon. (ºE) | Geomag. Lat. (ºN) | Geomag. Dip Lat. (ºN) | Modify Geomag. Dip Lat. (ºN) | Data Volume (Entries) |
|---|---|---|---|---|---|---|---|---|
| 1 | Akita | 39.70 | 140.10 | 205.33 | 29.01 | 53.67 | 46.88 | 6683 |
| 2 | Beijing | 40.00 | 116.30 | 184.60 | 28.04 | 57.28 | 48.80 | 8772 |
| 3 | Chongqing | 29.50 | 106.40 | 175.80 | 17.54 | 42.56 | 38.52 | 9551 |
| 4 | Guangzhou | 23.10 | 113.40 | 182.25 | 11.11 | 31.44 | 29.77 | 9239 |
| 5 | Haikou | 20.00 | 110.30 | 179.34 | 8.00 | 25.07 | 24.30 | 8213 |
| 6 | Irkutsk | 52.50 | 104.00 | 174.40 | 40.57 | 71.24 | 57.89 | 12,720 |
| 7 | Jeju | 33.50 | 126.50 | 193.90 | 21.89 | 47.86 | 42.45 | 1771 |
| 8 | Khabarovsk | 48.50 | 135.10 | 199.90 | 37.36 | 63.78 | 53.83 | 10,002 |
| 9 | Magadan | 60.00 | 151.00 | 210.09 | 50.13 | 71.46 | 60.45 | 8219 |
| 10 | Manzhouli | 49.60 | 117.50 | 185.32 | 37.66 | 67.45 | 55.64 | 8589 |
| 11 | Okinawa | 26.30 | 127.80 | 195.54 | 14.77 | 36.84 | 34.18 | 10,167 |
| 12 | Seoul | 37.20 | 126.60 | 193.74 | 25.59 | 52.73 | 45.88 | 3120 |
| 13 | Taipei | 25.00 | 121.50 | 189.77 | 13.19 | 35.01 | 32.70 | 10,824 |
| 14 | Tokyo | 35.70 | 139.50 | 205.31 | 24.99 | 48.91 | 43.45 | 15,046 |
| 15 | Tunguska | 61.60 | 90.00 | 164.56 | 50.18 | 78.38 | 63.25 | 9480 |
| 16 | Wakkanai | 45.40 | 141.70 | 205.89 | 34.82 | 59.61 | 51.15 | 15,271 |
| 17 | Yakutsk | 62.00 | 129.60 | 193.60 | 50.46 | 75.88 | 62.64 | 9840 |
| 18 | Yamagawa | 31.20 | 130.60 | 197.76 | 19.83 | 44.11 | 39.77 | 11,167 |

Data of different years are measured by different ionosondes. The early ionosondes are analog modulation with high cost, high transmitting power (up to 10 kW) and long observation time. While today's ionosondes are digital modulation with lower cost, lightweight, lower peak transmission power (less than 1000 W), wider frequency range (up to 30 MHz), various and self-customized frequency sweep resolution (such as 25 kHz, 50 kHz, 75 kHz or self-customized), greater flexibility (less than −120 dBm), easy to install and unattended "lights-off" remotely controlled operation with very little maintenance expense [38–41].

The measured database is spread across three ionospheric regions which cover high, middle and low latitudes (shown in Figure 2) and covers the years of 1949–2017 which are up to six solar activity cycles (shown in Figure 3). The time periods of 1949–2017 are used in the present study because it covers about six solar cycles as long as possible with the maximum data availability. Although not all the stations have data that are equally distributed within this period, the best use of the available data form each station is made. For the collected data, the following processing is done: (1) The sampling interval for 60 min was selected considering the periods of observations include 15, 30, and 60 min; (2) the monthly median values of $f_0F_2$ were calculated at each station where these values were not given in source dataset such as Okinawa, and Magadan station; (3) the stations with less than half solar activity period, such as Jeju station that has only data for the years of 2013–2017, are used for verification. Therefore, 17 stations excepting Jeju are used in temporal and spatial reconstruction. The ionosonde data for Manzhouli, Jeju and Haikou stations in the years of 2013 and 2017 were used for verification because they cover three ionospheric regions (high, middle and low latitudes) and covered two solar activity years (high and low solar activity years).

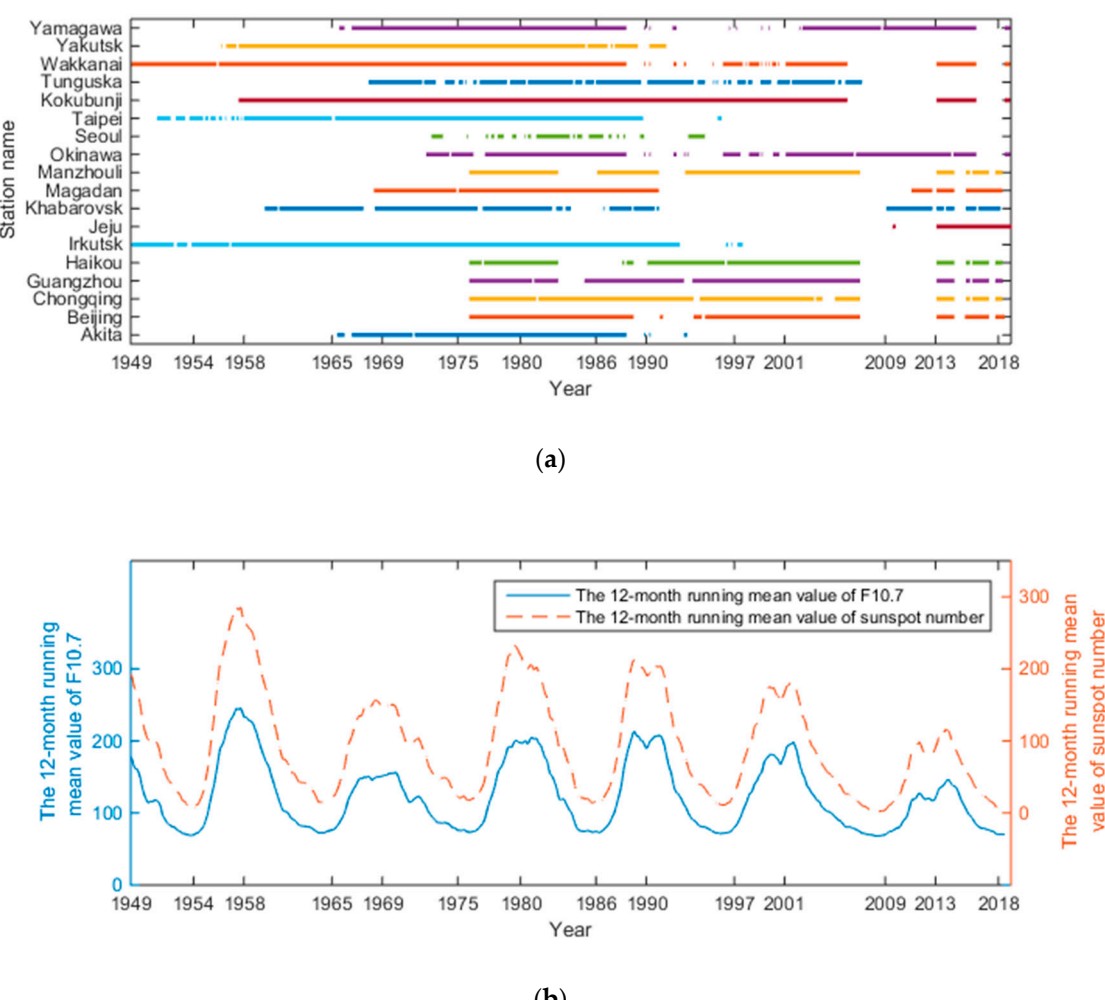

(**a**)

(**b**)

**Figure 3.** The time of data collected from ionosonde station and the corresponding cycle variations of solar activity factor ($F_{12}$ and $R_{12}$). (**a**) The time of data collected from ionosonde station; (**b**) the cycle variations of solar activity factor corresponding to the collected data.

## 3. Model Reconstruction

### 3.1. Temporal Characteristics Reconstruction

In describing solar activity, the flux of the solar radio waves at 10.7 cm ($F_{10.7}$) is affected by the corona and upper layer of the chromosphere, whereas sunspot activity is affected by the corona and the lower layer of the chromosphere [42]. A sample profile (shown in Figure 4) represents the correspondence between solar activity indices and monthly median values of $f_oF_2$ over Kokubunji station. There are monthly variations in the solar indices as well as monthly median values of $f_oF_2$ during the period from 1958 to 2005, covering four cycles in solar activity.

As shown in Figure 4, the similar cycle behaviors are reflected by $f_oF_2$ and solar indices. They respectively reached its peak and valley at the same corresponding years. At the same time, it is easy to see that the two solar activity parameters are not completely consistent with the variation of $f_oF_2$. In particular, the correlation coefficients between the $f_oF_2$ monthly median values and the twelve-month running mean value of solar indices at sample station (Kokubunji) is fully confirmed in Figure 5. The same conclusion has been reported in the literature [43]. As shown in Figure 5, the variation trends of the two types of correlation coefficients are generally similar but are different in some minor detail. During certain times and months, the correlation coefficients between $f_oF_2$ and the

two solar activity parameters are obviously different, especially at sunrise (around 06:00 clock) and sunset (around 20:00 clock) in summer.

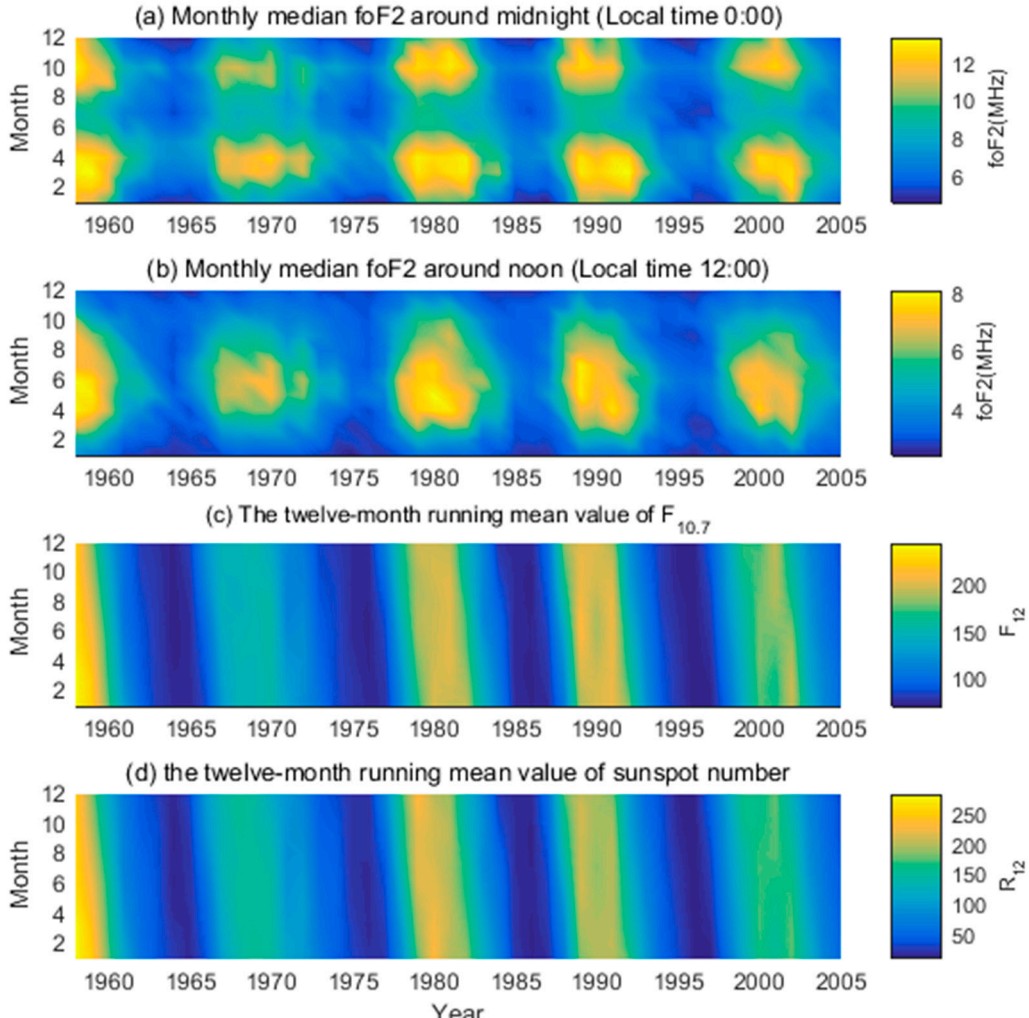

**Figure 4.** Behavior of the monthly median values of critical frequency and the twelve-month running mean value of solar indices at 4:00 UT during the years from 1958 to 2005 over Kokubunji, Japan. (**a**) Month median $f_oF_2$ (MHz) around midnight (Local time is 00:00 clock); (**b**) month median $f_oF_2$ (MHz) around noon (Local time is 12:00 clock); (**c**) the twelve-month running mean value of sunspot number ($F_{12}$); (**d**) the twelve-month running mean value of sunspot number ($R_{12}$).

Considering that the critical frequency is directly related to $F_{10.7}$ and the sunspot dissimilarity in different time periods, an alternative numerical map function is therefore derived. Concurrently, the annual and semi-annual variations of the ionosphere or more subtle variations are considered [44,45]. For given spatial location and time, $f_oF_2$ with the harmonic functions representing the seasonal (annual, semi-annual, quarter-annual, and monthly) and solar activity cycle variations (denoted by the variable $F_{12}, R_{12}, m$) can be formulated as the following:

$$
\begin{aligned}
\hat{f_o}F_2(F_{12}, R_{12}, m) &= \hat{f_o}F_2{}^F(F_{12}, m) + \hat{f_o}F_2{}^R(R_{12}, m) \\
&= \sum_{k=0}^{K} \sum_{l=0}^{L} \left[ c_{k,l} F_{12}{}^l \cdot \cos\left(\tfrac{2\pi km}{12}\right) + s_{k,l} F_{12}{}^l \cdot \sin\left(\tfrac{2\pi km}{12}\right) + \right. \\
&\quad \left. c'_{k,l} R_{12}{}^l \cdot \cos\left(\tfrac{2\pi km}{12}\right) + s'_{k,l} R_{12}{}^l \cdot \sin\left(\tfrac{2\pi km}{12}\right) \right]
\end{aligned}
\tag{4}
$$

where

$$\hat{f_o}F_2{}^F(F_{12}, m) = \sum_{k=0}^{K} \sum_{l=0}^{L} \left[ c_{k,l}F_{12}{}^l \cdot \cos\left(\frac{2\pi km}{12}\right) + s_{k,l}F_{12}{}^l \cdot \sin\left(\frac{2\pi km}{12}\right) \right], \tag{5}$$

$$\hat{f_o}F_2{}^R(R_{12}, m) = \sum_{k=0}^{K} \sum_{l=0}^{L} \left[ c'_{k,l}R_{12}{}^l \cdot \cos\left(\frac{2\pi km}{12}\right) + s'_{k,l}R_{12}{}^l \cdot \sin\left(\frac{2\pi km}{12}\right) \right], \tag{6}$$

and $c_{k,l}$, $s_{k,l}$, $c'_{k,l}$, and $s'_{k,l}$ are determined by the least square fitting approach. $K$ is the index representing the annual, semi-annual, seasonal and monthly cycle, $L$ is variations representing the solar cycle. The variations $K = 1, 2, 3$, and 4 of the trigonometric functions are expressed in terms of 12 months, 6 months, 3 months and 1 month respectively. The denominator 12 represent the maximum value of seasonal cycle, namely, twelve months. Moreover, the terms of solar activity can be expressed in terms of either a first-order, second-order, third-order or fourth-order function which is corresponding to $L = 1, 2, 3, 4$.

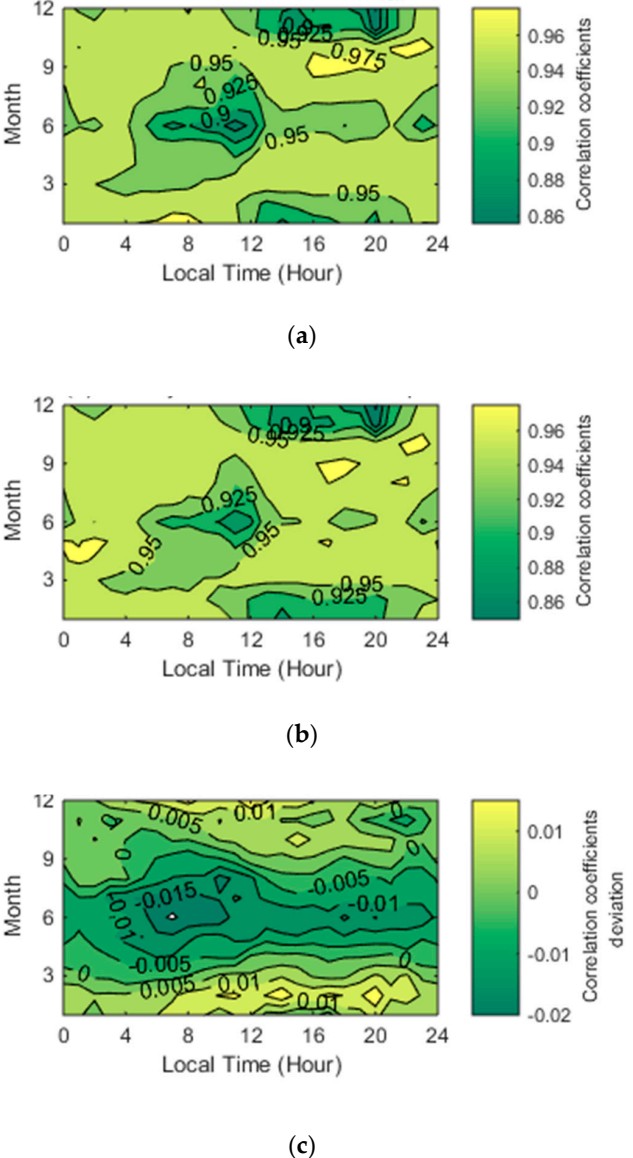

(a)

(b)

(c)

**Figure 5.** Correlation coefficients between the monthly median values of critical frequency and the twelve-month running mean value of solar indices over Kokubunji, Japan. (**a**) The correlation coefficients between monthly median $f_oF_2$ and $F_{12}$; (**b**) the correlation coefficients between monthly median $f_oF_2$ and $R_{12}$; (**c**) the deviation of two type correlation coefficients.

Figure 6 shows the root-mean-square error (RMSE) against different maximum regression order. As shown in Figure 6, it has better convergence and robustness with $L = 2$, which is better than that with $L = 1$. The root-mean-square error differences between measured value of $f_oF_2$ and regression values with $L = 2$ and $K = 2$ are already lower than that with $L = 1$ and $K = 4$. Furthermore, it is shown in Figure 6 that: the regress RMSE gradually decreased with considering the annual ($K = 1$), semi-annual ($K = 2$), seasonal ($K = 3$) and monthly ($K = 4$) cycle variation, when only using the maximum first-order form of solar activity parameter ($L = 1$). It is also noticed that the reconstruction results can obtain better convergence and better accuracy only considering the annual ($K = 1$) and semi-annual ($K = 2$) cycle variation, when using the maximum second-order of solar activity parameter ($L = 2$). Specially, the regression result based on the two solar activity index parameters is better than that based on the single parameter with 10.7 cm solar radio flux or sunspot. Considering the engineering simplification, the maximum orders of $K$ and $L$ was set as 2.

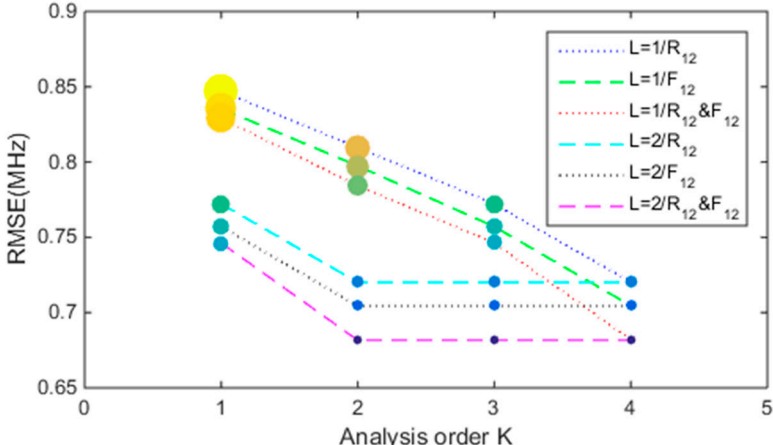

**Figure 6.** Bubble chart of Root-mean-square error of different analysis order. The size of bubble is set by $10^{7(REMS-0.5)}$ for better comparison viewing.

Figure 7 shows sample of regression coefficients at Kokubunji station. The label number from 1 to 36 represent the coefficients $c_{0,0}$, $c_{0,1}$, $c_{0,2}$, $s_{0,0}$, $s_{0,1}$, $s_{0,2}$, $c_{1,0}$, $c_{1,1}$, $c_{1,2}$, $s_{1,0}$, $s_{1,1}$, $s_{1,2}$, $c_{2,0}$, $c_{2,1}$, $c_{2,2}$, $s_{2,0}$, $s_{2,1}$, $s_{2,2}$, $c'_{0,0}$, $c'_{0,1}$, $c'_{0,2}$, $s'_{0,0}$, $s'_{0,1}$, $s'_{0,2}$, $c'_{1,0}$, $c'_{1,1}$, $c'_{1,2}$, $s'_{1,0}$, $s'_{1,1}$, $s'_{1,2}$, $c'_{2,0}$, $c'_{2,1}$, $c'_{2,2}$, $s'_{2,0}$, $s'_{2,1}$, and $s'_{2,2}$, respectively. As can be seen from Figure 7, the $c'_{0,0}$ is obviously more high than other coefficients in Kokubunji station. Moreover, the coefficients during the day are high than those during the night.

### 3.2. Spatial Characteristics Reconstruction

Ionospheric parameters such as $f_oF_2$ have an irregular non-homogeneous spatial distribution and therefore studies mapping its spatial characteristics have been conducted without interruption over several years. At present, numerous techniques based on different empirical or mathematical methods have been proposed to describe ionospheric spatial characteristics, including inverse distance weighted interpolation [46,47], spline interpolation methods [48–50], spherical harmonic (Legendre) functions [12,13], the SIRM [26,51,52] and improved versions [25] in Europe, along with statistical approaches such as kriging [53]. Among the above methods, inverse distance weighted interpolation is lack of geophysical mechanism and low accuracy where long distance from measured stations. The spherical Legendre functions recommended by the ITU [14,54] are only orthogonal over the whole sphere and do not efficiently represent a signal from a restricted region on the sphere [55].

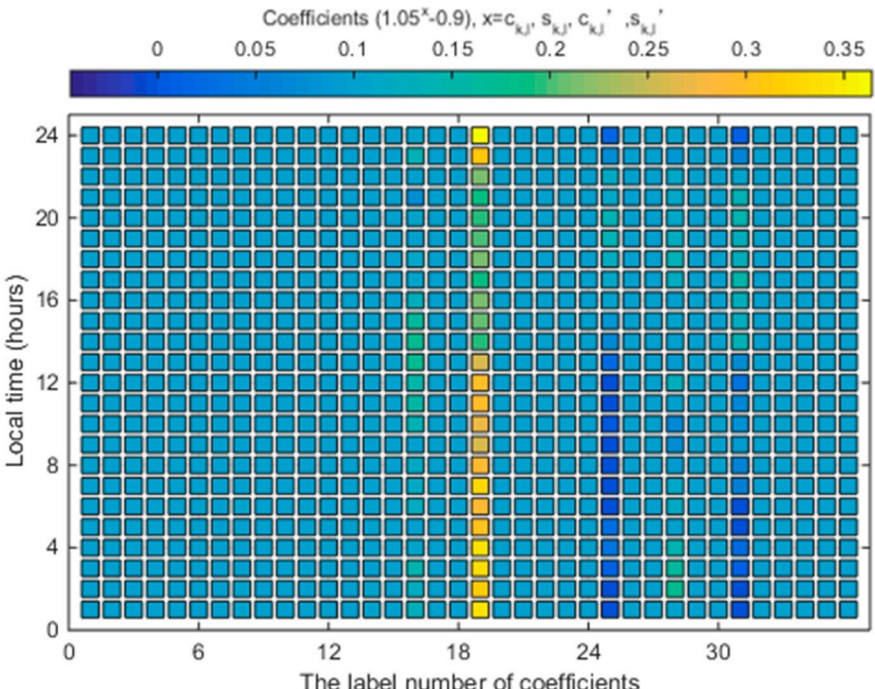

**Figure 7.** Sample of temporal reconstructed coefficients at 00:00 UT at Kokubunji station. The label number from 1 to 36 represent the coefficients $c_{0,0}$, $c_{0,1}$, $c_{0,2}$, $s_{0,0}$, $s_{0,1}$, $s_{0,2}$, $c_{1,0}$, $c_{1,1}$, $c_{1,2}$, $s_{1,0}$, $s_{1,1}$, $s_{1,2}$, $c_{2,0}$, $c_{2,1}$, $c_{2,2}$, $s_{2,0}$, $s_{2,1}$, $s_{2,2}$, $c'_{0,0}$, $c'_{0,1}$, $c'_{0,2}$, $s'_{0,0}$, $s'_{0,1}$, $s'_{0,2}$, $c'_{1,0}$, $c'_{1,1}$, $c'_{1,2}$, $s'_{1,0}$, $s'_{1,1}$, $s'_{1,2}$, $c'_{2,0}$, $c'_{2,1}$, $c'_{2,2}$, $s'_{2,0}$, $s'_{2,1}$, and $s'_{2,2}$, respectively.

On the other hand, surface spline interpolation based on radial basis functions (such as Green's functions) has excellent approximation properties and has become a mainstream method and a very powerful tool in multivariate approximations because of its high precision, simplicity, and flexibility [50,56]. Simultaneously, Green's functions have been tested and evaluated for analytical interpolation tests and shown to be the most robust, cost effective and accurate [57,58]. Furthermore, the Kriging method is an interpolation algorithm widely used in ionospheric parameter reconstruction and can offer the best linear unbiased estimates with an accurate description of the spatial structure of the ionospheric data. It uses known sample values and variogram to determine unknown values at different spatial locations. The variogram function of Kriging method describes the spatial correlation among the measured samples used in the interpolation and are calculated by taking the difference between pairs of measurements for a given distance [4,24,53]. Splines and kriging are two methods that should be used alternately, depending on what one wants to obtain [59].

To find the extreme accuracy limit of the regional model for $f_oF_2$, we implement two processes in developing an algorithm to describe the $f_oF_2$ spatial characteristics that provides high speed and high accuracy.

(1) Based on modified ionospheric distance from surface spline interpolation theory, the weights $W_n$ of equation (1) could be obtained from linear kriging equations:

$$\begin{cases} \sum_{j=1}^{N} r_{ij} \cdot w_j = r_{i0} - \mu, \ i = 1, 2, \cdots, N \\ \sum_{j=1}^{N} w_j = 1 \end{cases}, \tag{7}$$

where $N$ is maximum number of measured stations used to space interpolation, $\mu$ is the Lagrange factor, $r_{ij}$ is one of radial basis functions for modified ionospheric distance between two known measured

stations based on surface spline interpolation with modified radial basis function. It is expressed in matrix notation:

$$R = \begin{bmatrix} 0 & r_{12} & \cdots & r_{1N} \\ r_{21} & 0 & \cdots & r_{2N} \\ \vdots & \vdots & \ddots & \vdots \\ r_{N1} & r_{N2} & \cdots & 0 \end{bmatrix}, \tag{8}$$

On this condition, we can calculate the ionospheric distance vector between unknown location ($\vartheta_i$, $\phi_i$) and known measured station ($\vartheta_0$, $\phi_0$), which contains the semivariogram estimations ($R_0$), namely,

$$R_0 = \begin{bmatrix} r_{10} \\ r_{20} \\ \vdots \\ r_{N0} \end{bmatrix}, \tag{9}$$

Additionally, the weights $W$ of equation (1) could be obtained, which is expressed in vector notation:

$$W = \begin{bmatrix} w_1 \\ w_2 \\ \vdots \\ w_N \end{bmatrix}, \tag{10}$$

(2) In view of the facts that the ionosphere is controlled by both the orientation of the earth's rotation axis and the configuration of the geomagnetic field, the geomagnetic parameters play a significant role in mainly variations taking place in the ionosphere, most especially the F2 region. Simultaneously, the dependence of the tendency of changes in $f_oF_2$ on geomagnetic coordinate was proved [60]. Therefore, the geomagnetic functions of the geomagnetic longitude and dip latitude value are employed in the proposed model instead of simple geographic coordinates. Meanwhile, the modified radial basis function was defined in space reconstruction. Namely, the modified ionospheric distance between two different spatial locations in equation (7) is defined as following:

$$r_{ij} = \left[ \ln \left( \sqrt{(\phi_i - \phi_j)^2 + (\vartheta_i - \vartheta_j)^2} \right) - 1 \right] \cdot \left[ (\phi_i - \phi_j)^2 + (\vartheta_i - \vartheta_j)^2 \right]^2, \tag{11}$$

where $r_{ij}$ is the semivariogram value for the modified ionospheric distance between the *i*-th location ($\vartheta_i$, $\phi_i$) and *j*-th location ($\vartheta_j$, $\phi_j$), $\vartheta$ is geomagnetic latitude, $\phi$ is geomagnetic longitude.

In this study, we tested three types of geomagnetic coordinates magnetic Apex coordinates [61], corrected geomagnetic coordinates (CGM) [62], and modified geomagnetic dip latitude coordinates (MGD) [63,64]; see Figure 2. These coordinates are usually used in ionospheric physics and particularly suited for the organization of observations of plasma density in the ionospheric F2 layer [61]. From Figure 2, the trends for three types of coordinates obviously are different, and this circumstance becomes more serious for high-latitude regions.

To determine the coordinates, the ionosonde data from 17 modeling stations (except the Jeju station because its data only cover the period 2013–2017) were chosen of which Irkutsk, Manzhouli, Khabarovsk, Wakkanai, Beijing, Akita, Seoul, Kokubunji, Yamagawa, Chongqing, Okinawa, Taipei, Guangzhou, and Haikou were selected for generalized cross-validation. From the root-mean-square error (RMSE), Figure 8 similar trends appear for the four different coordinates, and the RMSEs for the stations in the mid-latitude region were lower than those in low and high-latitude regions. Specifically, the RMSEs of the geographic coordinates, magnetic apex coordinates, CGM coordinates, and MGD coordinates are 1.0272 MHz, 0.96895 MHz, 0.90360 and 0.95144 MHz. In particular, the RMSE using geomagnetic dip coordinates is seen to have a lower minimum than those using the other three types of

coordinates. The RMSE for the CGM coordinates is a minimum and lower than 0.1236 MHz (12.0327%) than for the geographic coordinates. Hence, the CGM coordinates were selected for the reconstruction of the spatial characteristics.

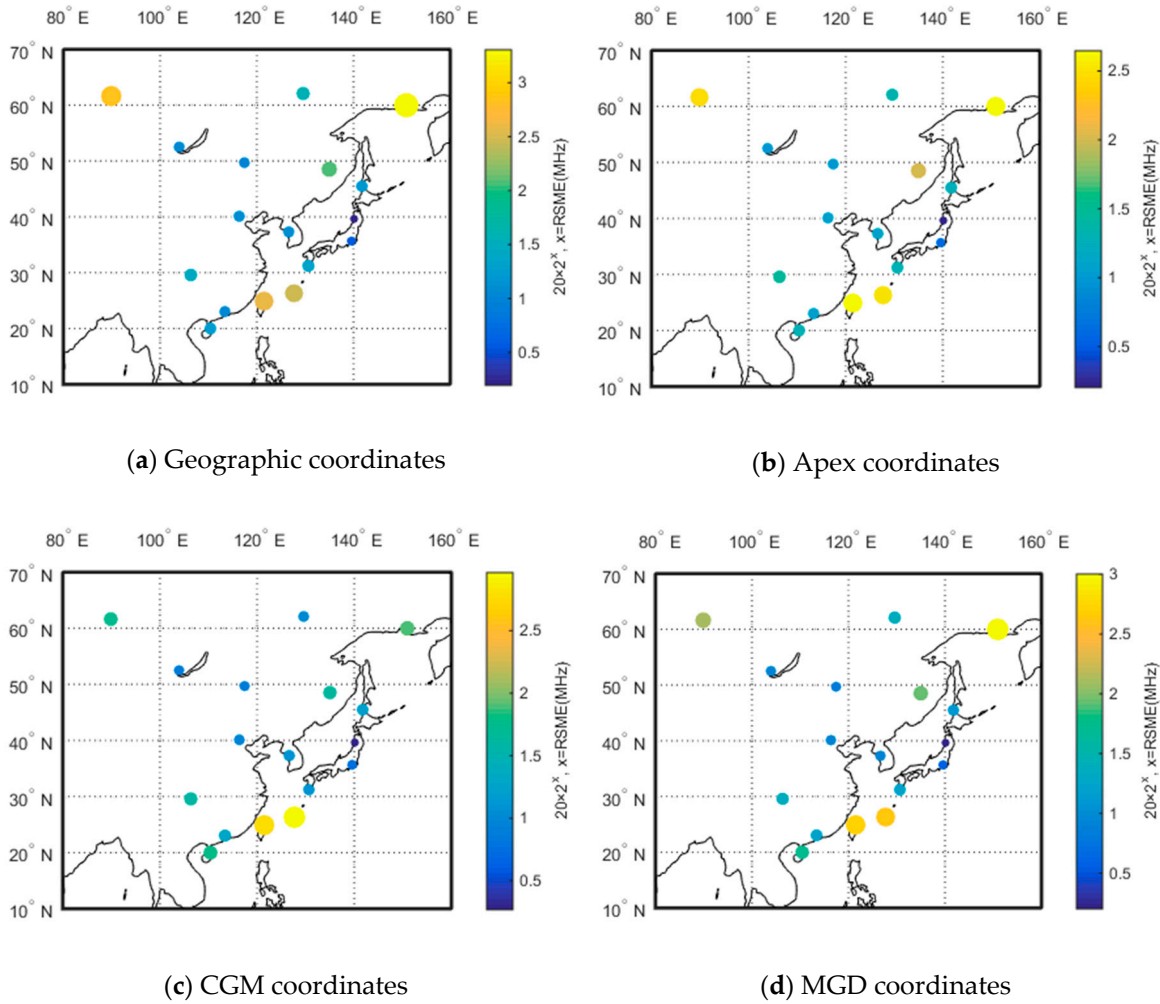

(**a**) Geographic coordinates

(**b**) Apex coordinates

(**c**) CGM coordinates

(**d**) MGD coordinates

**Figure 8.** Bubble chart of Root-mean-square error with four different coordinates. (**a**) Geographic coordinates, (**b**) Apex coordinates, (**c**) CGM coordinates, and (**d**) MGD coordinates. The radius of bubbles is set by $20 \times 2^{\text{RSME}}$ for better comparison.

Based on the above method, the map of weights *W* of equation (1) and (10) could be obtained, and the samples during four typical time period are shown in Figure 9. As shown in Figure 9, the weight variations are smooth with the solar activity parameters including the flux of solar radio waves at 10.7 cm and sunspot number. As well, there are different rules for different stations during different time period.

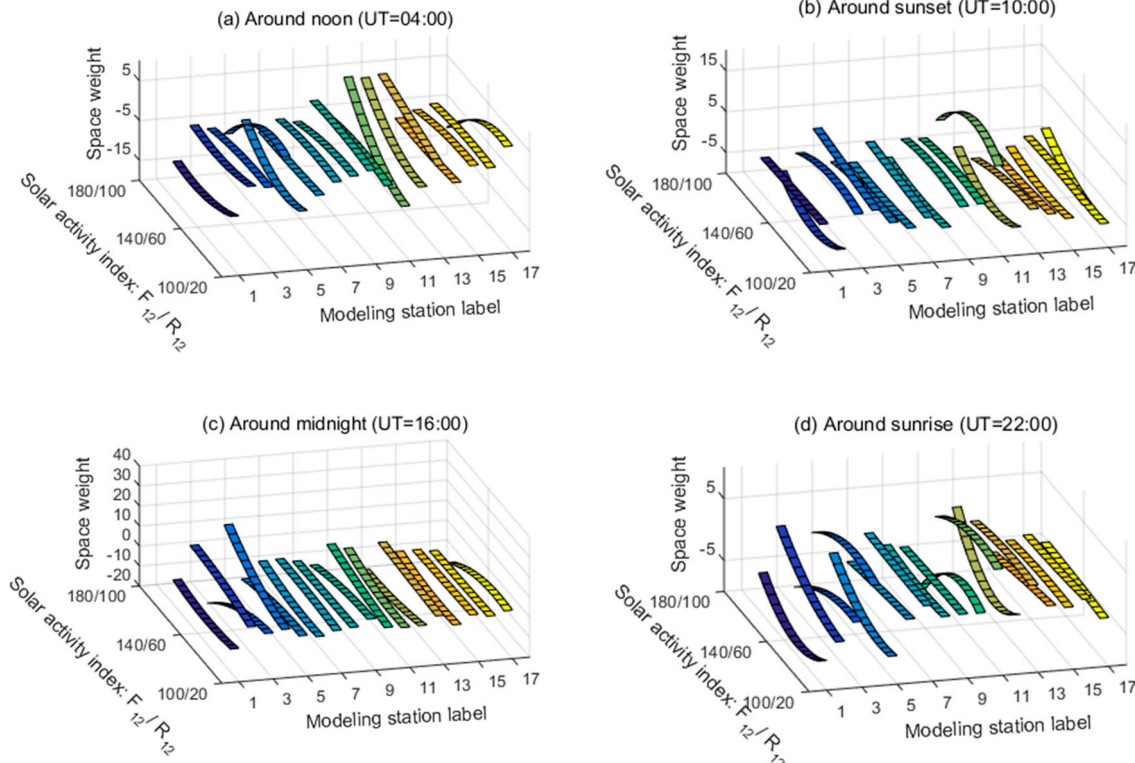

**Figure 9.** The sample maps of space weights with measured stations. Measured station labels from 1 to 17 are Akita, Beijing, Chongqing, Guangzhou, Haikou, Irkutsk, Khabarovsk, Magadan, Manzhouli, Okinawa, Seoul, Taipei, Tokyo, Tunguska, Wakkanai, Yakutsk, and Yamagawa, respectively. (**a**) Around noon (universal time is 00:00 clock); (**b**) around sunset (universal time is 10:00 clock); (**c**) around midnight (universal time is 16:00 clock); (**d**) around sunrise (universal time is 22:00 clock).

## 4. Discussions

To estimate and demonstrate the predictive capability of the proposed simplified regional prediction model (SRPM) concerning characteristic $f_oF_2$ values both the temporal (monthly and hourly) and spatial values of $f_oF_2$ predicted by the SRPM and those predicted by the IRI model (using both the URSI and CCIR coefficients) were compared with the observed values of $f_oF_2$ obtained from the verification stations (Figure 2) along with the RMSEs and the calculated relative differences (RRMSE) value. The latter two are used here to evaluate the performance from equations (2) and (3).

With the aforementioned equation and coefficients reconstructed using least-squares fitting regression, for given geographic coordinates ($\lambda$, $\varphi$) (where $\lambda$ and $\varphi$ refer to the geographical latitude and eastern longitude, respectively), month, universal time, and solar activity factor, we summarize the procedure for estimating $f_oF_2$ as follows: (1) Use the solar activity factor the 12-monthly mean 10.7 cm solar radio flux ($F_{12}$), the 12-monthly mean sunspot number ($R_{12}$), month (m), and universal time (UT) to calculate the temporal reconstruction variables for each modeling station. (2) Use the temporal relative variables and the geographic coordinates ($\lambda$, $\varphi$) to calculate the $f_oF_2$ monthly median value.

Figures 10 and 11 show some sample plots of the monthly mean values of $f_oF_2$ predicted values from the SRPM for the years of high solar activity (2013) and low solar activity (2017). They are compared with those determined by the IRI model using both the CCIR and URSI coefficients and measured values obtained from ionosonde stations. Among them, there is no data in Manzhouli and Haikou stations from April to August 2017. As can be seen from Figures 10 and 11, these graphs serve to illustrate that the SRPM and the IRI model successfully predict the general variation shape of behavior of $f_oF_2$. At the same time, it is found that the results also reflect the annual, semi-annual, seasonal, and monthly variation characteristics and the solar cycle variations of $f_oF_2$, which confirms

the validity of the modeling. From the whole aspect, the proposed model SRPM compared favorably with the IRI models.

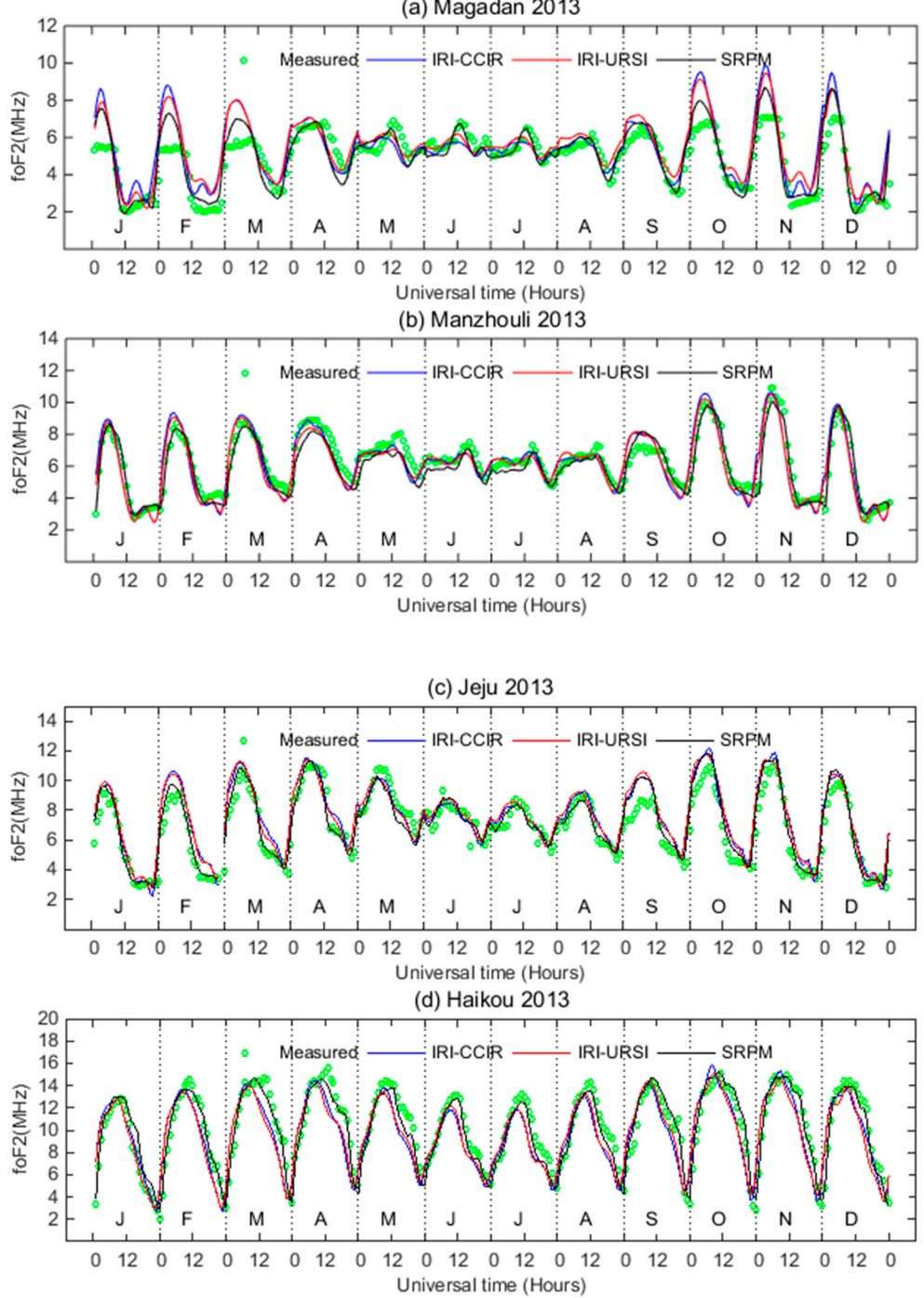

**Figure 10.** Sample plots of different model predicted value and measured value of $f_oF_2$ for the year 2013 (high solar activity year). From top to bottom panels, stations are ordered from high, middle and low latitude, which are Magadan, Manzhouli, Jeju and Haikou stations. (**a**) Magadan station; (**b**) Manzhouli station; (**c**) Jeju station; (**d**) Haikou station.

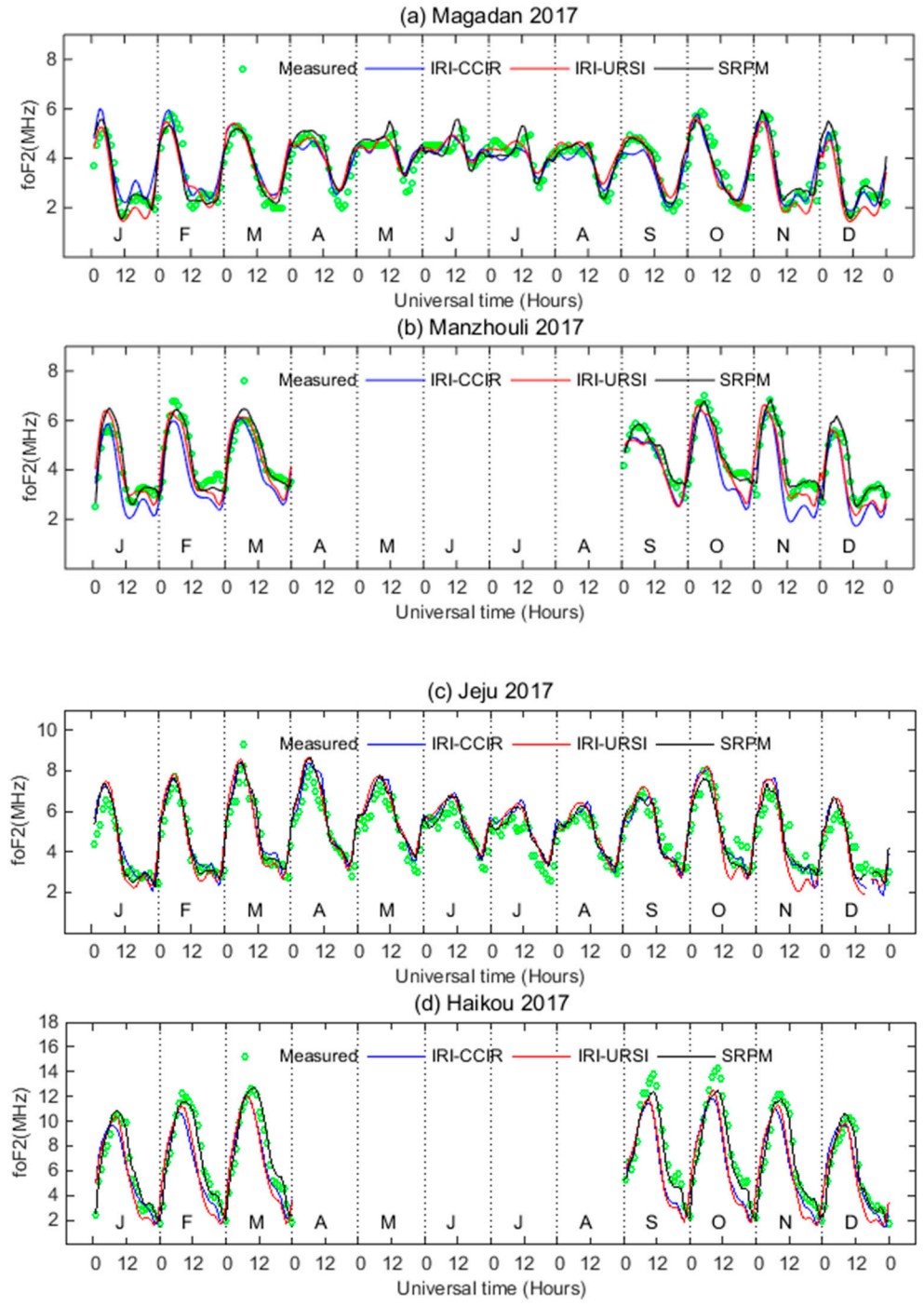

**Figure 11.** Sample plots of different model predicted value and measured value of $f_0F_2$ for the year 2017 (low solar activity year). From top to bottom panels, stations are ordered from high, middle and low latitude, which are Magadan, Manzhouli, Jeju and Haikou stations. (**a**) Magadan station; (**b**) Manzhouli station; (**c**) Jeju station; (**d**) Haikou station.

To estimate the accuracy of the proposed model SRPM, a statistical analysis of the differences between the predicted monthly mean values of $f_0F_2$ from the proposed model and measured values obtained from verifying stations is made by calculating RMSE and RRMSE. The results are compared with those determined by the IRI model using both the CCIR and URSI coefficients (see Figure 12 and Table 2). Figure 12 presents the RMSEs between measured values of $f_0F_2$ and predicted values of CCIR, URSI, and SRPM under different conditions, which correspond to two solar activity epochs (high and low), 12 months of the whole year, 24 h of the whole day and three latitudes regions (high, middle

and low). Table 2 lists the RMSE and RRMSE values for a comparison with the measured values of $f_oF_2$ and predicted values of CCIR, URSI, and SRPM under different conditions, corresponding to two solar activity epochs (high and low), the four seasonal periods (Spring, Summer, Autumn and Winter), four local time periods (midnight, sunrise, noon, and sunset), and three latitude regions (high, middle, and low).

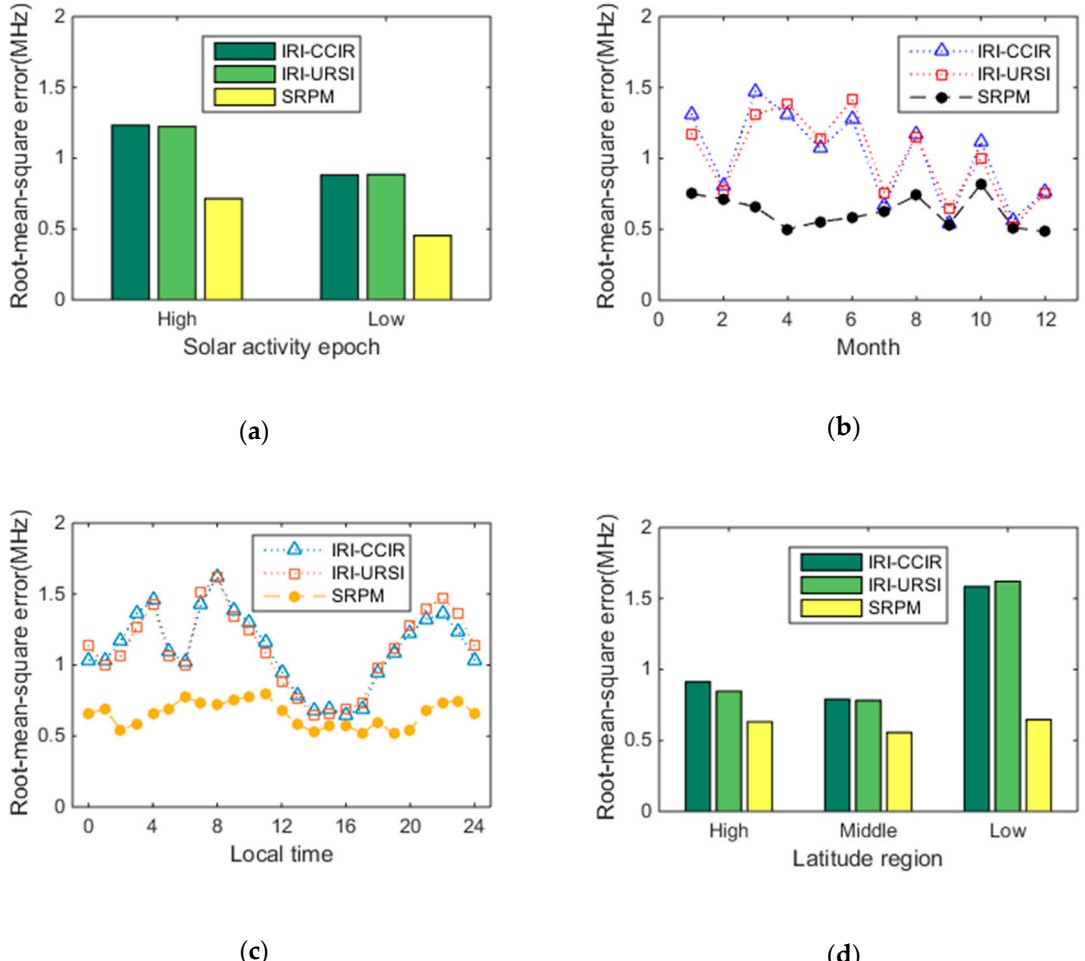

**Figure 12.** Root-mean-square error between measured value and predicted value of IRI model (using both URSI and CCIR coefficients) and SRPM. (**a**) Root-mean-square error of three prediction method during high and low solar epochs; (**b**) root-mean-square error of three prediction method in 12 months of the total year; (**c**) root-mean-square error of three prediction method at 24 h of the total day; (**d**) root-mean-square error of three prediction method over high, middle and low latitude regions.

These conclusions are drawn from Figure 12 and Table 2:

1. The SRPM is superior to CCIR and URSI models in both high and low activity years. As well, the overall percentage error difference between IRI and SRPM during high solar activity is higher than that during low solar activity;

2. Similarly, the SRPM is superior to CCIR and URSI models during four seasons. As well, the overall percentage error difference between IRI and SRPM is the maximum during spring and is the minimum during autumn. Specially, the variety trend of RMSE of CCIR is very similar to that of URSI during January to December as shown in Figure 12b;

3. It is easy to see that the SRPM is superior to CCIR and URSI models during different time sectors as shown in Figure 12c. Specially, the overall RMSEs during afternoon are minimum than other time sectors. There are similar for three models. As well the overall percentage error difference

from the test stations between IRI and SRPM is the maximum at the time sector of sunrise and is the minimum at the time sector of sunset;

4. The trend of RMSE from the test stations for CCIR are very similar to that for URSI during January to December as shown in Figure 12d. The RMSEs for CCIR and URSI in middle latitude region are the minimum than other regions, while the RMSEs for SRPM in three regions are very similar;

5. The average of RMSEs from the test stations for CCIR and URSI and SRPM are 1.082 MHz, 1.079 MHz, and 0.633 MHz respectively. As well, the average of the relative RMSE differences for CCIR and URSI and SRPM are 20.420%, 20.742%, 12.246% respectively. Especially, the SRPM model is better than the IRI model (CCIR and URSI) on average by a margin in the order of 8% (8.174% and 8.495%);

In summary, the predicted results from the proposed model SRPM performed better than those from the IRI model including both the CCIR and URSI. In other words, the SRPM improves on the IRI model.

**Table 2.** RMSE and relative value between measured values of $f_oF_2$ and predicted values of CCIR, URSI, and SRPM for two solar activity epochs, four seasons, four local time periods and three latitude regions.

| Statistical Analysis Item | | RMSE (MHz) | | | Relative RMSE (%) | | | Percent Difference between CCIR and SRPM (%) | Percent Difference between CCIR and SRPM (%) |
|---|---|---|---|---|---|---|---|---|---|
| | | IRI–CCIR | IRI–URSI | SRPM | IRI–CCIR | IRI–URSI | SRPM | | |
| solar activity epochs | high | 1.23 | 1.22 | 0.71 | 22.12 | 22.06 | 12.38 | 9.74 | 9.68 |
| | low | 0.88 | 0.89 | 0.45 | 19.04 | 20.02 | 11.22 | 7.82 | 8.80 |
| seasons | spring | 1.29 | 1.28 | 0.57 | 25.32 | 26.01 | 11.92 | 13.39 | 14.09 |
| | summer | 1.07 | 1.14 | 0.65 | 18.76 | 20.03 | 13.48 | 5.28 | 6.55 |
| | autumn | 0.79 | 0.75 | 0.64 | 12.05 | 11.88 | 11.14 | 0.91 | 0.74 |
| | winter | 0.99 | 0.92 | 0.66 | 24.97 | 23.46 | 14.77 | 10.20 | 8.69 |
| local time sectors | midnight (22:00–2:00) | 1.21 | 1.29 | 0.70 | 17.23 | 19.11 | 10.84 | 6.38 | 8.27 |
| | sunrise (5:00–9:00) | 1.34 | 1.35 | 0.72 | 33.84 | 33.72 | 18.69 | 15.15 | 15.03 |
| | noon (10:00–14:00) | 1.14 | 1.09 | 0.72 | 16.46 | 15.34 | 10.56 | 5.90 | 4.78 |
| | sunset (16:00–20:00) | 0.83 | 0.85 | 0.56 | 11.03 | 11.87 | 8.77 | 2.26 | 3.10 |
| latitude regions | high (≥60°N) | 0.91 | 0.85 | 0.63 | 24.21 | 23.43 | 16.27 | 7.93 | 7.16 |
| | middle (30°N–60°N) | 0.79 | 0.78 | 0.56 | 16.47 | 16.11 | 10.43 | 6.03 | 5.67 |
| | low (≤30°N) | 1.58 | 1.62 | 0.65 | 23.98 | 26.60 | 8.71 | 15.27 | 17.89 |
| average | | 1.08 | 1.08 | 0.63 | 20.42 | 20.74 | 12.25 | 8.17 | 8.49 |

## 5. Conclusions

A different empirical model of $f_oF_2$ based on an EOF and modified surface interpolation method was proposed. This model for predicting the ionospheric characteristics of the F2 layer instead of the CCIR/URSI coefficient was adopted in calculating $f_oF_2$. It has four main important features: (a) The ionospheric index $F_{10.7}$ and sunspot number were introduced together in the model to assist in the temporal reconstruction; (b) this model expresses $f_oF_2$ with harmonic functions representing the annual, semi-annual, seasonal, monthly and solar cycle variations; (c) the geomagnetic coordinates were applied instead of geographic coordinates for calculating ionosphere distance and using the spatial reconstruction; (d) this model has a very simple mathematical formulation that is more effective in representing temporal and spatial characteristics of $f_oF_2$.

The predicted results obtained from the SRPM agree well with the measurements. A statistical analysis has proved its potential in that it not only agrees well with the measurements but also performs better than the CCIR and URSI over East Asia region. To use the SRPM the new model requires only the following inputs: latitude, longitude, month, hour, the twelve-month running value of the monthly flux of solar radio waves at 10.7 cm, and the twelve-month running mean value of the monthly sunspot number. It is a simplified empirical model but has more accuracy than IRI. On the basis of the results obtained in this work, one may justify support this model for both regional and global $f_oF_2$ modeling. Future research is directed towards validating this modeling approach for other ionospheric

parameters such as propagation factor of 3000 km, and with modification confirming its suitability for other regions as well as globally.

**Author Contributions:** J.W. and J.M. came up with the idea and designed the proposed method; J.W., Q.C. and Y.C. collected and analyzed the data and simulation; J.W. and X.H. wrote this paper jointly; H.B., Q.C., Y.C. edited this paper; J.M. reviewed this paper.

**Funding:** This research was funded by National Natural Science Foundation of China (No. 61504092), National 973 Program of China (No. 61331901), and Qingdao National Laboratory for Marine Science and Technology of China (No. QNLM2016ORP0411).

**Conflicts of Interest:** The authors declare no conflict of interest.

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
