# Peer review of "Simplified Regional Prediction Model of Long-Term Trend for Critical Frequency of Ionospheric F2 Region over East Asia"

_applsci, doi:10.3390/app9163219_

Round 1
Reviewer 1 Report
Review of “Simplified Regional Prediction Model of long-term trend for critical frequency of ionospheric F2 region over East Asia” by Wang et al..
The manuscript describes an empirical model of the foF2 ionospheric characteristics over the East Asia sector. The SRPM (Simplified Regional Prediction Model) relies on a database of hourly foF2 values measured by East Asian ionosondes from 1949 to 2017 to model the long-term behavior of foF2 by jointly applying the Empirical Orthogonal Function analysis, the Fourier harmonic function theory and other spatial interpolation methods. The foF2 dependence on time has been described by an expansion on harmonic Fourier functions as a function of the month, the Universal Time, and on the solar activity indices R12 and F12. The spatial dependence has been described by a combination of the kriging method and surface spline interpolation method. The SRPM has been validated against foF2 values measured at 4 test stations, by showing better performances compared to the IRI model (by using both CCIR and URSI option for foF2 modeling).
The main scope of the manuscript is interesting for the ionospheric community and worthy of consideration. Overall, the manuscript is well written and logically organized. Anyhow, some shortcomings and inaccuracies came up by reviewing the paper. They need to be addressed before publication. This is why I propose a minor revision.
Below, I explicit my major and minor comments:
Major comments:
In the manuscript, the authors always write foF2, while the exact way of writing it is foF2 (only fo has to be written in Italic). This is a slight difference, but for this referee using the exact nomenclature is very important;
Lines 223-253: I have found this section very difficult to read and understand; furthermore, I think that are present some inaccuracies like: At line 223, you write ckj, skj, c’kj, and s’kj but the index j has never been specified. What is its meaning? Lines 225-228 are very difficult to understand. Please explain this part better; Line 230: why 30 coefficients? I would expect to have 64 coefficients because K=1,2,3,4 L=1,2,3,4 so 4x4=16 coefficients for each variable, so 16x4=64 coefficients. Please explain; Lines 239-250 are very difficult to understand. Please explain this part better.
Section 3.2: In this section you introduce so many spatial interpolation methods that eventually for the reader is very difficult to understand which of these you have used. Moreover, I cannot understand how you combined the kriging method with the surface spline interpolation based on modified radial basis function. This is not clear form the text. I suggest to rewrite this section making it more easily understandable for the reader;
Lines 387-443: This part is too much verbose. Values of RMSE and RRMSE for the different models and periods are already reported in Table 2, you do not need to rewrite them in the text. In this part, I would expect that you highlight main findings and differences between models, like done for example at lines 420-423. That would make the text lighter.
Minor comments:
Line 27: replace “that” with “those”; Line 38: replace “one” with “an”; Line 71: delete “getting”; Line 80: add a period between “[29]” and “The”; Lines 89-90: The sentence “It is controlled by the equatorial ionization anomaly” placed here can confuse the reader because the model embrace the entire East Asian sector and not only the low latitudes; Line 93: add a period between “[23,34]” and “Therefore”; Line 95: delete “around”; Line 97: delete “with”; Line 98: replace “is reconstructed with” with “are reconstructed through”; Line 99: replace “is reconstructed” with “are reconstructed”; Line 104: delete “in objection”; Line 121: add “the” before “geographic latitude” and before “geographic longitude”; Line 123: add “the” after “N is”; Line 124: replace “to interpolation” with “for interpolation”; Line 124: add “the” between “is” and “temporal”; Line 125: replace “is the weights” with “are the weights”; Line 142: replace “Figure 2, in where” with “Figure 2 and Table 1, where”; Line 144: replace “to model and verify” with “for model reconstruction and verification”; Line 146: delete “The” at the beginning of the line; Line 154: replace “was download” with “was downloaded”; Line 157: please specify if the number of ionosonde data refers to hourly values; Line 159: replace “station” with “stations”; Table 1: I think that the columns of the geographic longitude and latitude were erroneously switched; Line 161: delete “The” at the beginning of the line; Line 161: replace “ages are measured” with “years were measured”; Line 166: replace “less to” with “less than”; Labels of figure 2: replace “Vertification” with “Verification”, and “incluing” with “including”; Figure 2: It would be nice for the reader if you directly explained above each panel which geomagnetic coordinates are represented; Line 172: replace “are up six” with “are up to six”; Line 174: I think you are considering 6 and not 3 solar cycles as written; Line 180: replace “ station only has data” with “station that has only data”; Line 181: replace “only are used to be verified” with “are used for verification”; Lines 183-184: replace “covered” with “cover”; Figure 4: It would be nice to add, to the right of colorbars, the name of the represented physical quantity along with the unit of measure. Moreover, I would add the label “0 LT” and “12 LT” above the first two panels; Line 200: replace “midnight” with “noon”; Lines 202-203: The sentence “There are also… .. same years” is very difficult to understand, please correct; Line 209: replace “trivial” with “minor”; Line 209: The sentence “There are superior… .. and months” is very difficult to understand, please correct; Line 220: add commas between “annual”, “semi-annual”, and “monthly”; Line 223: replace “approaches” with “approach”; Lines 223-224: replace “is variations” with “is the index”; Line 233: replace “day from are high than” with “day are higher than”; Lines 243 and 292: “What’ more” ????? Line 284: It would be better to write “wn” instead of “W” here; Line 289: replace “calculated” with “calculate”; Equation (11); is the rij of equation (11) the same of equation (7)? It is not clear from the text; Figure 8: It would be nice to add, to the right of colorbars, the name of the represented physical quantity along with the unit of measure. Moreover, I would add a label above each panel describing the type of coordinates to which the panel refers; Line 324: add “the” between “on” and “above”; Line 324: replace “(9)” with “(10)”; Line 326: replace “smoothly” with “smooth”; Figure 9: I would add a label above each panel describing the LT to which the panel refers; Line 347: replace “stations” with “station”; Line 350: replace “showed” with “show”; Line 351: replace “are compared” with “compared”; Line 355: add “the” before “proposed” and “IRI”; Figure 10: I would add a label above each panel describing the station to which the panel refers; Line 360: replace “value” with “values”; Lines 360-361-364-365: replace “years of 2013” with “year 2013”; Figures 10-11 and lines 362-363-366-367: You write Hainan station but looking at Table 1 and Figure 2 I would expect to read Haikou station instead of Hainan. Please justify; Line 375: add a period between “low)” and “Table”; Line 386: replace “Those” with “These”; Line 438: replace “%” with “MHz”; Line 439: replace “Especially” with “Specifically”; Line 463: the reference [65] does not exist.

Reviewer 2 Report
Referee report on “Simplified Regional Prediction Model of Long-term Trend for Critical Frequency of Ionospheric F2 Region Over East Asia” by Wang et al. submitted to Applied Sciences
The authors present a new and superior predication model of regional long-term trends of the critical frequency in the ionospheric F2 region. The subject is of interest for the journal. The paper is well-written, the abstract presents a good summary, the title is appropriate, and the conclusion are supported by the context. The mathematical method is in general well-described. I only have some minor comments to an otherwise interesting and valuable paper to the community. Hence in my opinion the paper can be published in Applied Sciences after minor amendments detailed below:
Line 35: Could the authors state that that critical frequency is the plasma frequency and give the basic formula to avoid confusion with other critical frequencies, mainly for the sake of non-specialists.
Line 128: should be ”…comparison between measured values and different models will…”
Line 141: The authors appear to be using 18 ionosonde stations, rather than 15. Please clarify.
Line 141-142: grammar bug. Possibly delete “where” in line 142.
Line 157: Clarify what the number 168674 is measured in, which is described as “data volume”. Entries? Bytes?
Line 161: Clarify units of the last column of the table as in the previous comment.
Line 172: Should be “….which are up to six…”
Line 199: Also connected with my first comment, please state explicitly if you give the frequency or the angular frequency w=2pi f. It would help if you gave a formula when you define the critical frequency.
Line 220: should be “…annual, semi-annual…”
Line 221: In equation 4, could you elaborate explicitly what m/12 represents.
Line 243: Replace “What’ more” by “Additionally, …” or similar.
Line 243: Should be “…regression…”
Line 266: should be “…offer…”
Line 288: should be “…stations…”
Line 289: should “…we can calculate…”
Line 292: Replace “What’ more” by “Additionally, …” or similar.
Line 326: should be “…are smoothly related with…” or similar.
Line 350: Should be “…show…”. Some problem with grammar in this sentence. E.g. in line 351, one could start a new sentence: “….model SRPM. They are compared…”
